# Endurance Exercise Training Mitigates Diastolic Dysfunction in Diabetic Mice Independent of Phosphorylation of Ulk1 at S555

**DOI:** 10.3390/ijms25010633

**Published:** 2024-01-03

**Authors:** Yuntian Guan, Mei Zhang, Christie Lacy, Soham Shah, Frederick H. Epstein, Zhen Yan

**Affiliations:** 1Fralin Biomedical Research Institute, Center for Exercise Medicine Research at Virginia Tech Carilion, Roanoke, VA 24016, USA; yg6ju@virginia.edu (Y.G.); christie21@vtc.vt.edu (C.L.); 2Center for Skeletal Muscle Research at Robert M. Berne Cardiovascular Research Center, University of Virginia, Charlottesville, VA 22903, USA; 3Departments of Pharmacology, School of Medicine, University of Virginia, Charlottesville, VA 22903, USA; 4Departments of Medicine, School of Medicine, University of Virginia, Charlottesville, VA 22903, USA; 5Departments of Biomedical Engineering, School of Medicine, University of Virginia, Charlottesville, VA 22903, USAfhe6b@virginia.edu (F.H.E.); 6Departments of Molecular Physiology and Biological Physics, School of Medicine, University of Virginia, Charlottesville, VA 22903, USA; 7Department of Human Nutrition, Foods, and Exercise, College of Agriculture and Life Sciences, Virginia Tech, Blacksburg, VA 24061, USA

**Keywords:** diabetes, diastolic dysfunction, exercise intervention, mitophagy, mitochondrial quality, echocardiography

## Abstract

Millions of diabetic patients suffer from cardiovascular complications. One of the earliest signs of diabetic complications in the heart is diastolic dysfunction. Regular exercise is a highly effective preventive/therapeutic intervention against diastolic dysfunction in diabetes, but the underlying mechanism(s) remain poorly understood. Studies have shown that the accumulation of damaged or dysfunctional mitochondria in the myocardium is at the center of this pathology. Here, we employed a mouse model of diabetes to test the hypothesis that endurance exercise training mitigates diastolic dysfunction by promoting cardiac mitophagy (the clearance of mitochondria via autophagy) via S555 phosphorylation of Ulk1. High-fat diet (HFD) feeding and streptozotocin (STZ) injection in mice led to reduced endurance capacity, impaired diastolic function, increased myocardial oxidative stress, and compromised mitochondrial structure and function, which were all ameliorated by 6 weeks of voluntary wheel running. Using CRISPR/Cas9-mediated gene editing, we generated non-phosphorylatable Ulk1 (S555A) mutant mice and showed the requirement of p-Ulk1at S555 for exercise-induced mitophagy in the myocardium. However, diabetic Ulk1 (S555A) mice retained the benefits of exercise intervention. We conclude that endurance exercise training mitigates diabetes-induced diastolic dysfunction independent of Ulk1 phosphorylation at S555.

## 1. Introduction

Diabetes mellitus (DM) affects over 422 million people worldwide and is growing at an alarming rate, with the patient population being projected to be over 693 million by 2045 [1,2]. It is well documented that cardiovascular complications in diabetic patients are a significant and deadly comorbidity [3,4]. In particular, patients with diabetes suffer from a significantly increased risk of developing heart failure even in the absence of common causes, such as coronary artery diseases, hypertension, and atherosclerosis [5,6]. An early clinical manifestation of heart failure in a diabetic patient is reduced left ventricular relaxation during the diastolic phase, defined as diastolic dysfunction. Diabetes-induced diastolic dysfunction has become an important research topic in recent years. It is likely that numerous possible mechanisms underlying this disease condition were suggested. The main cellular defects responsible for diabetes-induced diastolic dysfunction are still largely unknown, greatly impeding therapeutic developments. 

Autophagy is an intrinsic, physiological catabolic pathway that removes dysfunctional or damaged cytoplasmic components, including damaged/dysfunctional mitochondria. During autophagy, the “cargo” is recognized and engulfed by double-membrane structures termed autophagosomes, which are then fused with lysosomes for lysosomal degradation. A highly conserved form of autophagy that specifically targets damaged/dysfunctional mitochondria is defined as mitophagy, which is a critical step of mitochondrial quality control [7]. In the myocardium, autophagy and mitophagy are particularly important as cardiomyocytes, the main functional cells that undergo continuous contractions, require efficient cellular clearance to deal with various cellular and energetic stresses [8,9]. A growing number of studies have found that these cellular clearance pathways are impaired in cardiac pathologies, including diabetes [10,11,12,13]; however, their precise roles in cardiovascular diseases remain elusive. For example, while excessive cardiac autophagy might be detrimental in causing cell death and exaggerating cardiac insults, exercise training also promotes mitophagic activity and provides protection against heart failure [14,15,16,17]. Therefore, it is of great interest to investigate the roles and regulations of myocardial autophagy/mitophagy in response to exercise training under the condition of diabetes.

Regular exercise is the most effective, non-invasive, physiological, and cost-effective intervention for diabetic complications, and endurance exercise training is proven to improve diastolic function in diabetic patients [18,19,20]. The impacts of exercise on diastolic dysfunction in diabetes are pluripotent and multi-systemic. Importantly, the underlying mechanism(s) remains poorly understood. One potential mechanism of exercise benefits is enhanced mitophagy [21,22]. This postulation is supported by numerous recent studies suggesting that diabetes-induced functional impairment of the heart may stem from loss of mitochondrial respiratory function and/or accumulation of damaged or dysfunctional mitochondria [23,24,25]. Previous studies from our lab have shown that endurance exercise promotes mitophagy to restore metabolic disturbances in skeletal muscle [26]. Since the myocardium is highly dependent on mitochondrial respiratory function for contractility, we therefore postulate that endurance exercise training mitigates diabetes-induced diastolic dysfunction by promoting mitophagy. 

Unc-51-like kinase 1 (Ulk1, also known as ATG1) is a highly conserved kinase that was first discovered as an important driver of autophagy initiation; recent literature has shown that Ulk1 is essential for cardiac mitophagy [27,28]. Animal studies with genetic and pharmacological inhibitions showed that Ulk1-mediated mitophagy is activated and important during functional recovery in mouse models of cardiac insults, including diet-induced obesity and surgical injury of ischemia/reperfusion [28,29,30]. Interestingly, Ulk1 is activated when serine 555 is phosphorylated by 5′ AMP-activated kinase (AMPK), a master energy-sensing kinase for energetic stress and exercise cues [31,32]. We have recently shown that mice with skeletal muscle-specific knockout of Ulk1 had a loss of exercise-induced mitophagy in skeletal muscle and exercise-mediated improvement of insulin signaling [26]. However, the role of Ulk1 activation through serine 555 phosphorylation in exercise benefits has never been investigated in the diabetic heart. 

Here, we employed a murine model of severe diabetes with a combination of high-fat diet (HFD) feeding and streptozotocin (STZ) injections that induce exercise intolerance and diastolic dysfunction. Voluntary wheel running (6 weeks) mitigated diastolic dysfunction and endurance intolerance in the diabetic mice along with significant improvements in mitochondrial structure and respiratory function in the myocardium. Using CRISPR/Cas9-mediated gene editing, we generated mice with a mutation of serine 555 in Ulk1 to non-phosphorylatable alanine (S555A). The mutant mice expressed non-activatable Ulk1 and had impaired mitophagy induced by acute exercise. Interestingly, when we subjected diabetic Ulk1 (S555A) mice to long-term voluntary running, the benefits of exercise were retained.

## 2. Results

### 2.1. A Combination of High-Fat Diet (HFD) and Streptozotocin (STZ) Injections Induces Severe Diabetes in Mice

An established mouse model of diabetes using HFD and STZ was employed (Figure 1A) [33,34,35,36]. HFD-fed mice gained significant body weight before STZ injections compared to control mice, which then was reduced to the same levels as control mice following STZ injections (Figure 1B). Despite the metabolic stress from HFD and STZ injections, DM-Ex mice achieved a daily running distance of approximately 7 km (Figure 1C). Importantly, all diabetic mice (DM-Sed and DM-Ex) had severe hyperglycemia and glucose intolerance compared to control mice. These parameters were not improved after exercise training in the DM-Ex group likely due to the destruction of the insulin-secreting pancreatic β cells by STZ injections (Figure 1D). 

### 2.2. Diabetic Mice Develop Exercise Intolerance, Which Is Mitigated by Voluntary Wheel Running

At the end of the 14-week protocol, an exhaustive treadmill running test with an increment speed regimen and inclination (5%) was performed for each mouse to assess the endurance capacity and overall cardiovascular fitness according to our previous studies [26,37]. DM-Sed mice showed significantly decreased running capacity compared to control mice (Figure 1E). Physical exhaustion during the treadmill running test was validated by elevated blood lactate at the end of the running session (Figure 1E). Exercise capacity was significantly improved by 6 weeks of voluntary wheel running in the DM-Ex group (Figure 1E). These findings provide strong evidence that endurance exercise training mitigates exercise intolerance in severely diabetic mice. 

### 2.3. Diabetic Mice Have Preserved Systolic Function but Impaired Diastolic Function That Are Alleviated by Voluntary Wheel Running

To assess cardiac function, cardiac magnetic resonance imaging (MRI), with Displacement Encoding with Stimulated Echoes (DENSE) and dobutamine stress echocardiogram, was performed. At rest, both DM-Sed and DM-Ex mice showed normal left ventricle (LV) strain via DENSE imaging (Figure 2A), suggesting normal systolic function in these mice. A dobutamine stress echocardiogram evaluates cardiac function after β-adrenergic agonism (bolus injection of dobutamine at 2.5 µg/g body weight, i.p.). As shown in Figure 2B,C, both DM-Sed and DM-Ex mice had normal responsiveness in increasing ejection fraction (EF) and heart rate (HR) in response to dobutamine. Although both DM-Sed and DM-Ex mice had a trend of reduced LVESD compared to control mice, the differences were not statistically significant (Figure 2D). Preserved EF and LV end-systolic diameter (LVESD) in the diabetic mice (regardless of exercise training) indicated systolic function was preserved, consistent with the MRI-DENSE results (Figure 2A). However, during diastole, the DM-Sed group had a significantly decreased ability to refill the left ventricle, indicated by a significantly decreased LV end-diastolic diameter (LVEDD) post-dobutamine injection (Figure 2E). Intriguingly, this diastolic dysfunction was almost completely mitigated by 6 weeks of voluntary running. Taken together, mice with severe diabetes and significant cardiometabolic stress to the myocardium have diastolic dysfunction when tested with a dobutamine stress echocardiogram, and exercise training mitigated diastolic dysfunction even in the absence of improved glycemic control. Notably, DM-Sed mice had a reduced heart weight when normalized to tibia length, which was also restored by voluntary wheel running (Figure 2F). These findings suggest that severe diabetes induced by HFD and STZ injections leads to cardiac atrophy, which can be mitigated by endurance exercise training. Cardiac fibrosis, as a result of adverse accumulation and structural remodeling of myocardial fibrillar collagen matrix expression, has been suggested as a major cause of diastolic dysfunction. In the present study, both DM-Sed and DM-Ex showed no significant increase of fibrotic tissue assessed by picro-sirius red staining (Appendix A) and western blot analyses of fibrosis markers (Appendix A). These findings indicate that the loss of diastolic function in our mouse model of severe diabetes is not due to cardiac fibrosis and that endurance exercise training exerts its benefits through other mechanism(s).

Transmission electron microscopy (TEM) was performed to evaluate the mitochondrial content and morphology in the left ventricle (LV) tissues of all groups. Mitochondrial content, size, circularity, and cristae quality were analyzed. Quantification of mitochondria by TEM did not show any significant changes in LV mitochondrial content (Figure 3A–E), which was confirmed by western blot analyses of mitochondrial electron transport chain proteins (Figure 3I). However, LVs of DM-Sed mice showed increased mitochondria with a lower cristae density (defined less than 50% of occupancy of clearly defined cristae structure in a mitochondrion), a higher degree of mitochondrial fragmentation (significantly more mitochondria with small cross-sectional area), and a more irregular shape based on circularity score (Figure 3A–E). Exercise training alleviated the cristae and size abnormality but not shape irregularity (Figure 3A–E). The oxygen consumption rate (OCR) of isolated cardiac mitochondria with pyruvate and malate as substrates was measured to evaluate mitochondrial respiration capacity (Figure 3F). DM-Sed mice showed a significantly reduced state III (stimulated by 5 mM ADP) and FCCP-induced respiration compared to control mice, which were completely restored by exercise training (Figure 3G). DM-Sed mice also exhibited heightened levels of oxidative stress as indicated by increased protein carbonylation with a clear trend of reduction toward the level of control mice following 6 weeks of voluntary running (Figure 3H).

### 2.4. Phosphorylation of Ulk1 at S555A Is Required for Exercise-Induced Mitophagy in the Heart

To study the necessity of the phosphorylation of ULK1 at S555 (an activation site by AMPK), we generated Ulk1 knock-in (KI) mice with a mutation of serine 555 to alanine. The genotypes of the Ulk1-S555A mice were confirmed by sequencing of tail DNA and western blot analysis of the heart tissues (Figure 4A). We then subjected the KI mice and the wildtype littermates to an acute bout of treadmill running (90 min), followed by 6 h of recovery, to examine the role of p-Ulk1 at S555 in exercise-induced mitophagy by measuring mitophagy markers in isolated mitochondria via western blot. An acute bout of treadmill running resulted in an increased abundance of light-chain 3-II (LC3-II) in the mitochondria of wild-type littermates (Figure 4B). *Ulk1-S555A* mice had elevated LC3-II levels in mitochondria at baseline (without exercise), but the increases of LC3-II by acute exercise were completely ablated in the KI mice (Figure 4B). The elevated mitochondrial LC3-II in KI mice at baseline could be due to increased mitochondrial damage and/or impaired mitophagy execution. The ablation of exercise-induced increase of mitochondria-associated LC3-II suggests that Ulk1 phosphorylation at S555 is required for exercise-induced mitophagy in the heart.

### 2.5. Ulk1-S555A Knock-In Mice Retain the Benefits of Exercise Training in Improving Exercise Capacity under the Condition of Severe Diabetes

We next sought to investigate whether Ulk1 activation-mediated mitophagy is critical for the exercise benefits in the diabetic heart. We subjected *Ulk1-S555A* KI mice and the wildtype littermates to the same diabetes and exercise protocol as described earlier. The body weight of *Ulk1-S555A* KI mice showed a similar trend following HFD feeding and STZ administration, while KI mice without an HFD and STZ had a slightly higher baseline body weight (Appendix A). The glucose tolerance test revealed that KI mice, under the condition of HFD/STZ treatment, had the same degree of hyperglycemia and impairment of glycemic control as their wildtype littermates (Figure 4C). Both WT and KI diabetic mice ran approximately the same daily distance during voluntary running training (Figure 4D). Interestingly, exercise intolerance in KI diabetic mice was completely mitigated by wheel running, indicating that the phosphorylation of Ulk1 at S555, presumably Ulk1-mediated mitophagy, is not required for exercise-mediated mitigation of exercise intolerance (Figure 4E,F).

### 2.6. Ulk1-S555A Mice Retain Exercise-Mediated Protection against Diastolic Dysfunction under the Condition of Severe Diabetes

A dobutamine stress echocardiogram was performed to evaluate the cardiac function and the impact of voluntary running. All six groups of mice responded to dobutamine with a similarly increased ejection fraction (Figure 5A,B). Both diabetic sedentary groups (WT-DM-Sed and KI-DM-Sed) displayed a significantly greater reduction of diastolic diameter after dobutamine injection, which was mitigated by 6 weeks of voluntary running (Figure 5C,D). These findings suggest that *Ulk1-S555A* knock-in mice had similar diastolic dysfunction with preserved systolic function compared to the wildtype littermates under the condition of severe diabetes. Voluntary wheel running mitigated diastolic dysfunction in both the wildtype and knock-in groups, indicating that phosphorylated Ulk1 at S555 is not required for exercise-mediated protection against diastolic dysfunction in this mouse model of diabetes.

### 2.7. Ulk1-S555A Mice Retain Similar Mitochondrial Adaptations of Exercise in Severe Diabetes

TEM images were acquired to assess LV mitochondrial morphology. Similar to the wildtype cohort, *Ulk1-S555A* knock-in mice showed a higher degree of mitochondrial fragmentation indicated by significantly increased small mitochondria (<0.4 µm^2^) (Figure 6A–C). The circularity score in the diabetic KI mice was not significantly different from the control KI mice, possibly due to a smaller sample size. Exercise intervention alleviated the impaired size abnormality in both WT and KI mice (Figure 6B). Mitochondrial OCR was performed to assess mitochondrial respiration in the KI mice, as mentioned above. The KI-DM-Sed mice had a significantly reduced state III (ADP) respiration, while in the KI-DM-Ex mitochondria, it was slightly improved and not significantly different from the healthy control (Figure 6D). These data suggest that Ulk1 phosphorylation at S555 is not required for mitochondrial adaptation of exercise in the diabetic model. Additionally, we performed western blot analyses of BCL2 and adenovirus E1B 19-kDa-interacting protein 3 (Bnip3), which is a mitophagy adapter with an unclear regulatory mechanism [38,39]. We found a significantly increased expression of Bnip3 in the DM-Sed group of both genotypes, while exercise intervention resulted in a further significant increase regardless of genotype (Figure 6E). The changes in the diabetic heart were accompanied by a decrease in the abundance of cardiac 3-hydroxybutyrate dehydrogenase 1 (Bdh1), a key protein mitochondrial ketone body metabolism, in both WT and KI mice (Figure 6E).

## 3. Discussion

In this study, we employed an established diabetes model in mice by an HFD-STZ protocol where the diabetic mice showed diastolic dysfunction in response to β-adrenergic stimulation while systolic function was preserved. This phenotype was associated with impaired mitochondrial morphology and respiratory function in the left ventricle. Importantly, 6 weeks of exercise intervention in the form of voluntary wheel running mitigated diastolic dysfunction with significantly improved mitochondrial morphology and respiration as well as the alleviation of oxidative stress. To dissect the molecular mechanism of the benefits of exercise, we generated Ulk1-S555A knock-in mice and subjected these mice to the same diabetes condition with/without exercise interventions. Our findings suggest that endurance exercise training is highly efficacious in mitigating diastolic dysfunction and associated pathological changes in mice with severe diabetes, and that the exercise benefits are not dependent on S555 phosphorylation (activation) of Ulk1.

The employment of an exercise intervention in our study is of great clinical relevance. It is widely accepted that aerobic exercise training is beneficial for diabetic complications, especially those who show an early onset of diastolic dysfunction. For example, Hollekim-strand and colleagues reported in a randomized trial that at-home high-intensity interval exercise over 12 weeks effectively improved diastolic function in patients with type II diabetes in both genders [18]. However, adherence to exercise intervention has been a major issue due to exercise intolerance [40,41,42,43,44]. In the present study, the diabetic mice showed this important clinical feature of exercise intolerance that was completely alleviated by exercise training even under the condition of continued severe hyperglycemia, suggesting an insulin sensitivity-independent mechanism of exercise benefits.

It is important to emphasize that the present model of diabetes has several limitations. First, a relatively short duration of diabetes in mice (14 weeks) may not fully recapitulate the clinical cardiovascular complications in humans. Since diabetic patients have approximately a four-fold higher risk of developing heart failure when they are diagnosed for more than 15 years [45], cardiovascular complications could be a result of an accretion of long-term exposure to metabolic perturbations in the heart. For example, we found no clear evidence of fibrosis and hypertrophy (Appendix A) in the current model, whereas these abnormalities may be prevalent in diabetic patient populations. The current model may represent an early, mild impairment of cardiac function. Second, the duration of hyperglycemia is short (5 days of STZ injections) and acute (fasting glucose at over 400 mg/dL and increased to over 600 mg/dL during GTT) compared to humans, which may underscore the metabolic differences between mice and humans. STZ is also not the normal cause of diabetes, which might induce off-target effects. Studies have used high doses of STZ to induce type one diabetes (>200 mg/kg), which may cause detrimental whole-body effects [46]. In the present study, we chose to use low-dose, consecutive injections to avoid the detrimental effects. We observed changes that are consistent with the literature. Although we think that this model is justifiable in the present study because the functional benefits of exercise are clear, we urge caution when interpreting these findings concerning clinical conditions.

Taking advantage of mouse genetic models, we sought to dissect the role of mitophagy on exercise benefits in diabetic hearts. As shown by acute exercise experiments, the phosphorylation of Ulk1 at S555 appeared to be important for the baseline level of mitophagy, as indicated by an elevated LC3-II in the mitochondrial fraction (Figure 4B). Meanwhile, acute exercise-induced mitophagy, as indicated by an increased LC3-II in the mitochondrial fraction 6h following exercise recovery, was impaired in the Ulk1-S555A mice (Figure 4B). Our findings refuted our hypothesis and support that *Ulk1-S555A* mice are equally protected from exercise training (Figure 5). There are at least two alternative mechanisms. First, mitophagy may be regulated by redundant cellular machinery. For example, loss of Bnip3 has recently been correlated with dilated cardiomyopathy and impaired mitophagy via unknown mechanisms [38,39]. Here, we detected diabetes-induced expression of Bnip3 with a further increase by exercise intervention in both wildtype and Ulk1-S555A mice (Figure 6E). These findings may suggest alternative mitophagy pathways after chronic exercise (as opposed to a single bout of acute stress) that are independent of Ulk1. Another possibility is the PTEN-induced kinase 1 (PINK1) and Parkin mitochondrial membrane potential monitoring system that has been associated with cardioprotective potentials [47,48,49]. We have recently reported that a pool of AMPK on the outer membrane of mitochondria senses mitochondrial energetics and contributes to mitophagy induction [50]. It is entirely possible that other mitophagy receptors are at play in regulating mitophagy by sensing mitochondrial energetic status. Second, there may be other adaptive responses in the heart to endurance exercise that are sufficient to confer exercise benefits. For example, endurance exercise has been shown to promote eNOS-dependent mitochondrial biogenesis in mouse hearts [51]. Overall, these adaptive changes may overcome the deficit of mitochondrial respiratory function and confer functional benefits in the Ulk1-S555A mice. 

At the heart of diastolic dysfunction is the dysregulation of muscle contractility and, therefore, exercise training may have positive impacts on this important component of the cardiac cycle. Numerous studies have shown that the function of sarco/endoplasmic reticulum Ca^2+^ ATPase 2a (SERCA2a), a main driver of diastole, is compromised in the diabetic myocardium [52,53,54,55], and the stimulation of SERCA2a has been shown to have a significant positive impact on diastolic function in diabetic mice [55]. Our original hypothesis entails that excessive ROS production may disrupt both the regulation and function of SERCA2a, which could be mitigated by exercise-mediated mitophagy with improved mitochondrial quality and reduced ROS production [56,57]. Our finding of reduced protein carbonylation by exercise training is consistent with this notion; however, more definitive evidence remains to be obtained. Exercise training may also promote the expression of SERCA2a in the heart [58]. It is also important to note that exercise training may result in improved contractility in the diabetic heart that may not be dependent on improved mitochondrial quality. For example, exercise training can lead to increased cytoplasmic Ca^2+^ sensitivity in cardiomyocytes potentially driven by increased intracellular pH, which could be a contributing factor in the exercise-mediated protection of diastolic dysfunction in a rat model [59]. Future studies should elucidate these exercise effects to gain more in-depth mechanistic insights.

Several other limitations of the present study need to be considered when interpreting the data. Cardiac energy metabolism is significantly changed in the context of diabetes and diastolic dysfunction [60,61]. Studies have found that diabetes causes an increased reliance on fatty acid metabolism and decreased glucose metabolism. In the present study, mitochondrial oxygen consumption assays were performed with sodium pyruvate and L-malic acid as substrates (Figure 4 and Figure 6). These experiments aimed to assess the respiratory capacity of isolated mitochondria, and we have not addressed the effects of altered metabolism and its role in the Ulk1-S555A mice. Second, cardiac ketone metabolism has been found as a defense mechanism against cardiac injuries, such as heart failure. Studies have confirmed a protective function of enhanced expression of 3-hydroxybutyrate dehydrogenase 1 (Bdh1), a key mediator of ketone body metabolism in the heart [62,63]. We found that, in diabetic hearts, Bdh1 is significantly downregulated, agreeing with similar findings in the field [64,65]. Exercise intervention in this particular model did not mitigate this impairment in both WT and Ulk1-S555A mice, suggesting the existence of different mechanisms of exercise benefits other than enhanced Bdh1 expression. Future investigations should incorporate more delicate dissections of mitochondrial substrate utilization, such as in the presence of fatty acid-related metabolic substrates. Metabolomics studies are also warranted to provide more metabolic insight into diabetes and exercise benefits. 

In conclusion, we have shown that a combination of an HFD and STZ causes severe diabetes with exercise intolerance, diastolic dysfunction, and compromised mitochondrial quality as well as increased oxidative stress in the heart. These abnormalities were all mitigated by 6 weeks of voluntary wheel running in the absence of improved hyperglycemic control. A mutation of Ulk1 at serine 555 to alanine disrupts acute exercise-induced mitophagy but does not lead to loss of exercise benefits in the diabetic heart. These findings suggest that endurance exercise training mitigates diabetes-induced diastolic dysfunction independent of Ulk1 phosphorylation at S555.

## 4. Materials and Methods

### 4.1. Animals

All animal procedures were approved by the Institutional Animal Care and Use Committee at the University of Virginia (Approval code: 3762. Approval date: 2 August 2022). Wild-type male C57BL/6 mice (12–13 weeks old) were obtained commercially from Jackson Laboratories (Farmington, CT, USA). *Ulk1-S555A* knock-in mice (in C57BL/6 background) were generated at the Genetically Engineered Murine Model (GEMM) core facility at the University of Virginia. All mice were housed in 2–4 mice/cage except for voluntary running mice, which were single-housed, in temperature-controlled (21 °C) quarters with a 12:12-h light-dark cycle and ad libitum access to water and normal chow (Purina). To induce diabetes in mice (DM), male mice of 12–13 weeks of age were subjected to high-fat diet (HFD, 60% kcal from fat, Research Diets, New Brunswick, NJ, USA) feeding. After 4 weeks of HFD feeding, all mice received 5 daily intraperitoneal (i.p.) injections of streptozotocin (STZ, Sigma, Burlington, MA, USA) at 50 mg/kg (in citrate buffer, pH = 4.0) followed by three weekly doses of STZ at 20 mg/kg. At week 8, blood glucose was measured using tail vein blood to confirm hyperglycemia (>250 mg/dL). Half of the DM mice were subjected to voluntary wheel running for 6 weeks (DM-Ex), and their daily running activities were recorded, while the rest of the DM mice were subjected to sedentary cage activities (DM-Sed). Another group of sedentary mice fed on normal chow without STZ injection, however, vehicle buffer (citrate buffer) served as control (Con). 

### 4.2. Glucose Tolerance Test (GTT)

Glucose tolerance tests were performed at the end of the diabetes exercise protocol as described previously [26,37,66]. All mice were placed in regular cages and fasted for 6 h starting at 9:00 am. After taking a measure of fasting blood glucose by a glucometer (Ascensia Bayer, Basel, Switzerland) from the tail vein, a bolus of D-glucose dissolved in normal saline was injected at 2 g/kg body weight (i.p.). Tail vein blood glucose levels were measured at 30, 60, and 120 min post glucose injection. 

### 4.3. Dobutamine-Stress Echocardiography

This test was performed with a slightly modified protocol from a well-established method at least 48 h after the GTT test [67]. Briefly, mice were maintained on anesthesia via an inhalation mask using 2.5% isoflurane mixed with O_2_ at a flow rate of 200 mL/min in a supine position on a warm plate maintained at 37 °C. Two-dimensional-guided M-Mode images were acquired in short-axis view with a 13 MHz linear transducer before and at 0, 5, 10, and 15 min after dobutamine (Sigma, USA) injection (2.5 µg/g body weight, i.p.). All images were analyzed with FIJI software (version 1.53f, NIH, Bethesda, MD, USA). 

### 4.4. Treadmill Running Test

Exercise capacity was assessed by an exhaustive treadmill running test for all mice according to our previous studies [26,37]. Briefly, at least 48 h after dobutamine-stress echocardiography, all mice were familiarized with the treadmill with 3 daily running sessions at 0° incline and 13.41 m/min for 10 min (day 1–3). On day 4, mice were subjected to running on a fixed 5% incline at increment speed which increased by 3 m/min every 30 min, with a starting speed of 13.41 m/min (0.5 mph), until exhaustion. Mice were stimulated by manual prodding with a hard brush placed inside the treadmill lane. Perceived exhaustion was confirmed with pre- and post-running blood lactate measurements from the tail vein. 

### 4.5. Cardiac DENSE-MRI Analysis

Cine displacement encoding with stimulated echoes magnetic resonance imaging (DENSE-MRI) was assessed for all groups according to a previously established protocol [68,69,70]. Imaging was performed on a midventricular short-axis slice with the following parameters: field-of-view = 30 × 30 mm^2^, matrix size = 128 × 128, slice thickness = 1 mm, flip angle = 15 degrees, echo time = 0.67 ms, repetition time = 7.1 ms, number of averages = 4, number of spiral interleaves = 82, and displacement encoding frequency = 0.4 cycles/mm. Peak circumferential end-systolic strain was calculated using the DENSE analysis tool [69]. 

### 4.6. Tissue Preparation

Tissue harvesting was conducted after mice were euthanized under 2.5% isoflurane at least 48 h after the last test. Mice were fasted for 16 h (overnight) before tissue collection. Heart tissues were removed with surgical tools, rinsed in 1X PBS, weighed, and allocated to different assay preparations. For TEM analyses, a ~2 mm piece of left ventricular tissue was fixed at 4 °C overnight in 4% paraformaldehyde and 2.5% glutaraldehyde in 0.1 M NaCacodylate. A portion of the heart was embedded and frozen with O.C.T. compound submerged in freezing isopentane (Fisher Scientific, Waltham, MA, USA) for cryostat processing. Heart sections were sliced at 10 µm in O.C.T. using a Leica CM1860 cryostat (Leica Biosystems, Richmond, IL, USA). Tissue lysates for protein analyses were homogenized in glass homogenizers with protein loading buffer containing 50 mM Tris-HCl, pH 7.4, 1% sodium dodecyl sulfate (SDS), 10% glycerol, 20 mM dithiothreitol, 127 mM 2-mercaptoethanol, and 0.01% bromophenol blue, supplemented with protease inhibitors (Roche) and phosphatase inhibitors (Sigma-Aldrich). Homogenized lysates were boiled in a heat block at 97 °C for 4 min before western blotting or frozen in −70 °C.

### 4.7. Western Blotting

Protein lysates were subjected to sodium dodecyl sulfate-polyacrylamide gel electrophoresis and wet transferred onto a nitrocellulose membrane. Membranes were probed with the following primary antibodies at a 1:1000 dilution: targeting S555 p-ULK1 (CST #5869), S757 p-ULK1 (CST #6888), ULK1 (Sigma-Aldrich #A7481), Vdac (CST #4866), OXPHOS cocktail (Abcam #110413), and Gapdh (CST #2118). The secondary antibodies were goat anti-mouse IR680 and goat anti-rabbit IR800 (LICOR). Membranes were scanned using the Odyssey infrared imaging system (LI-COR, Lincoln, NE, USA). Proteins were analyzed in comparison to a common protein standard loaded on the gel, prior to calculating phospho:total, signal:Ponceau, or protein:Gapdh ratio.

### 4.8. Mitochondrial Oxygen Consumption Rate (OCR) Assay

Cardiac mitochondria for OCR assays were isolated via centrifugation from heart tissues homogenized by an electric grinder in isolation buffer containing 0.5% BSA, 70 mM sucrose, 210 mM mannitol, 5 mM HEPES, and 1 mM EGTA (pH 7.2) at 4 °C. To evaluate mitochondrial OCR, fractionated mitochondrial pellets were gently resuspended in respiration assay solution containing 0.2% BSA, 70 mM sucrose, 220 mM mannitol, 10 mM KH_2_PO_4_, 10 mM MgCl_2_, 2 mM HEPES, 1 mM EGTA, 10 mM sodium pyruvate, and 2 mM L-malic acid (pH 7.2). Protein concentration was determined by Bradford protein assay (Thermo-Fischer, Agawam, MA, USA), and mitochondrial samples were temporarily stored on ice. A total of 25 µg of mitochondria was then added to a pre-warmed (37 °C) optical O_2_ sensing system (Instech Laboratories, Plymouth Meeting, PA, USA) in duplicates for the OCR assay. The O_2_ concentration was continuously recorded in 3 min intervals after different reagents were added in the following order: no reagent (state II), 500 mM ADP (state III), 5 µM oligomycin, and 7 µM carbonyl cyanide-p-trifluoromethoxyphenylhydrazone (FCCP). The slope of oxygen consumption was analyzed in Prism 9 (Graphpad, Boston, MA, USA) from the last minute before the addition of the next reagent (or until OCR became linear). The O_2_ chambers were thoroughly washed with 0.5% Triton-100X followed by ample deionized water to avoid cross-contamination of samples and reagents in between experiments. 

### 4.9. Statistical Analyses

Data are presented as mean ± SEM. Experiment results in the C57BL/6 mice with 3 groups were analyzed via one-way ANOVA with Newman-Keuls post-hoc analysis. In the *Ulk1-S555A* cohort where 6 groups were present, data was analyzed using two-way ANOVA with Bonferroni post-hoc analysis where appropriate. Where only one variable was present, data was analyzed using Student’s *t*-test. Statistical significance was established a priori as *p* < 0.05.

## Figures and Tables

**Figure 1 ijms-25-00633-f001:**
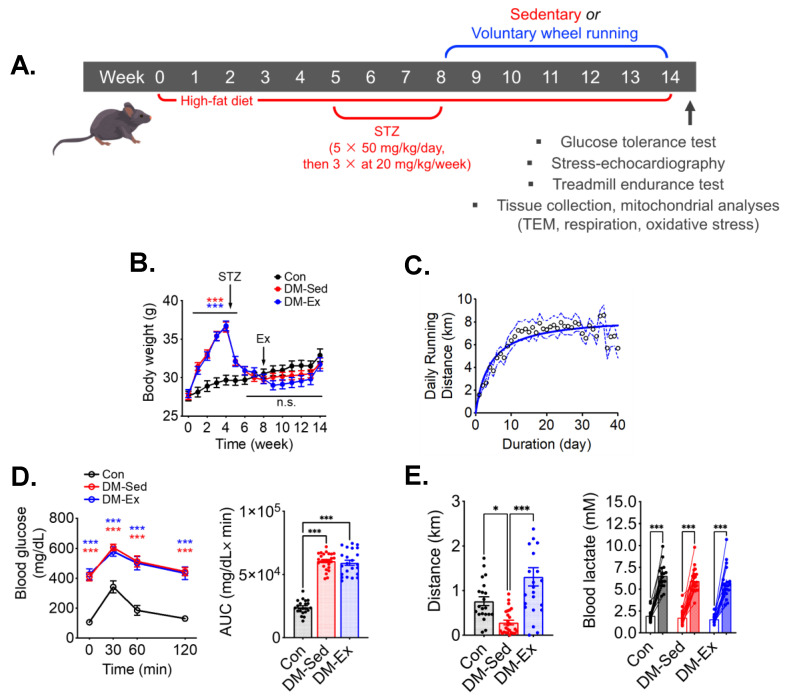
A combination of HFD and STZ injections induces severe diabetes with exercise intolerance in mice. (**A**) Protocol for diabetes induction and exercise intervention. STZ, streptozotocin. (**B**) Body weight during the course of study. Ex, the start of exercise intervention. n.s., not significant. *n* = 18, 23, 21 for Con, DM-Sed, and DM-Ex, respectively. (**C**) Daily running distance in the DM-Ex group. Dotted line, the 95% confidence intervals. (**D**) Left, blood glucose during GTT and area under the curve (AUC) taken at the end of GTT. ***: *p* < 0.0001 compared with Con (one-way ANOVA with post hoc multiple comparisons). Some high data points were substituted with 650 mg/dL due to surpassing the glucometer limit. (**E**) Left, endurance capacity test result. *n* = 18, 23, 21 for Con, DM-Sed, and DM-Ex, respectively; *: *p* < 0.05, ***: *p* < 0.001 from one-way ANOVA with post hoc multiple comparisons. Right, blood lactate was performed before and after the exhaustive running session to confirm perceived exhaustion. ***: *p* < 0.001, paired *t*-tests of individual pre- and post-running session.

**Figure 2 ijms-25-00633-f002:**
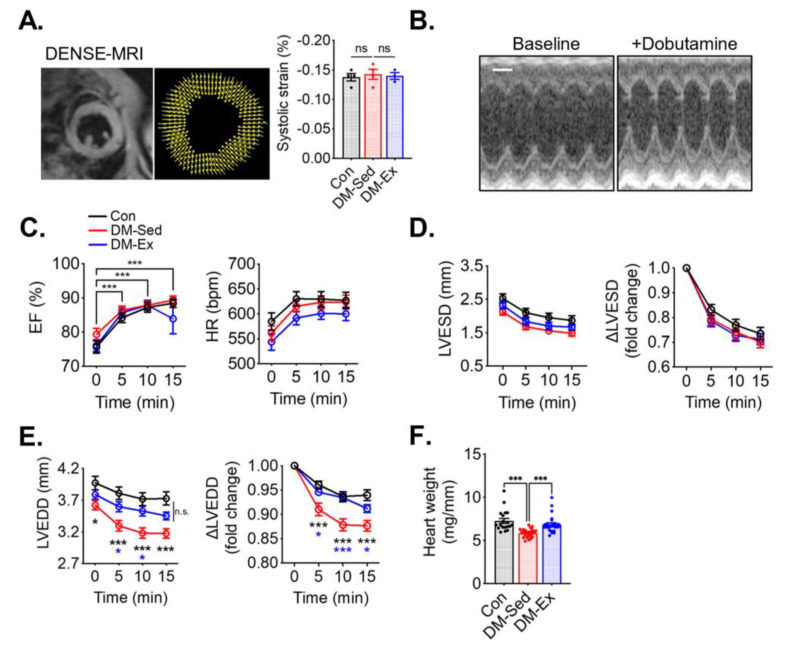
Exercise training mitigates diastolic dysfunction and cardiac atrophy in diabetic mice. (**A**) Example and results of LV systolic strain analysis from cardiac magnetic resonance imaging (MRI) with Displacement Encoding with Stimulated Echoes (DENSE) imaging. n.s., not significant. *n* = 4, 4, 3 for Con, DM-Sed, and DM-Ex groups, respectively. (**B**) Dobutamine stress echocardiogram. Representative images from a control mouse at baseline and 10 min post-i.p. injection of dobutamine (2.5 µg/g). scale bar, 1 mm. (**C**) Ejection fraction (EF) and heart rate (HR) in the dobutamine stress echocardiogram. ***, *p* < 0.001 compared with baseline from one-way ANOVA with post hoc multiple comparisons at respective time points. (**D**,**E**) Quantifications of LV end-systolic diameter (LVESD) and LV end-diastolic diameter (LVEDD). *, *p* < 0.05, ***: *p* < 0.001 compared with baseline in the same group, one-way ANOVA with post hoc multiple comparisons. (**F**) Heart weight (mg) normalized by tibial length (mm); *n* = 18, 24, 26 for Con, DM-Sed, and DM-Ex groups, respectively, ***: *p* < 0.001, one-way ANOVA with post hoc multiple comparisons.

**Figure 3 ijms-25-00633-f003:**
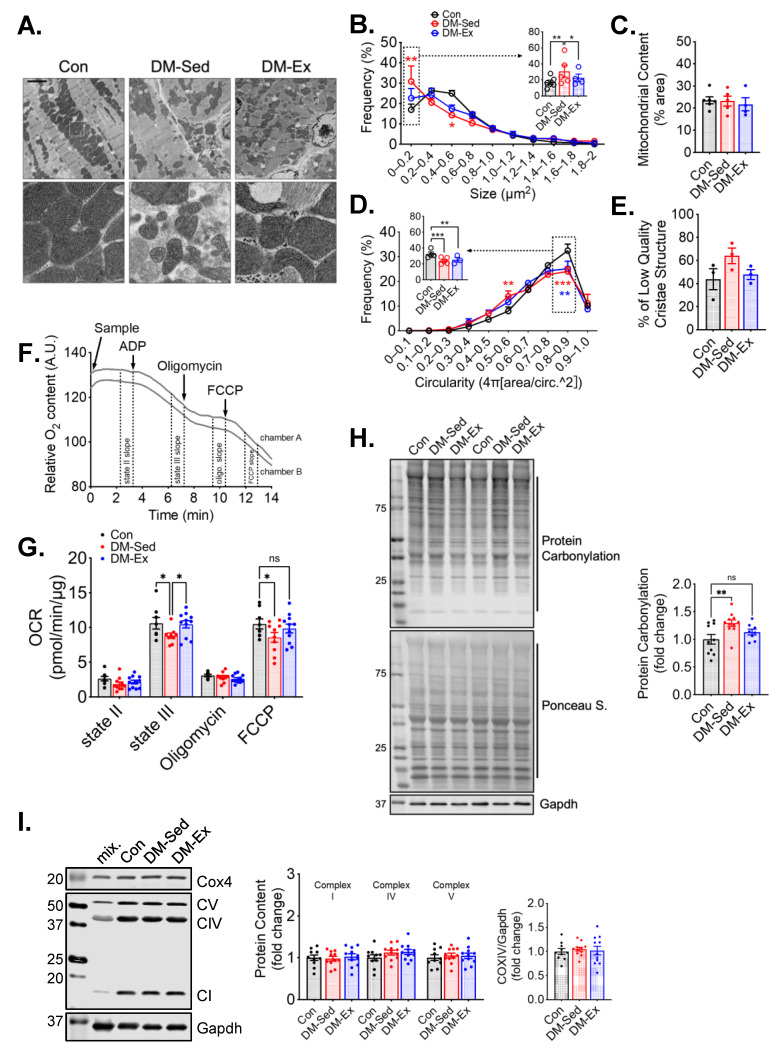
Exercise training mitigates mitochondrial impairments in the left ventricle without altering mitochondrial content. (**A**) Representative transmission electron microscopy (TEM) images of LV. Scale bar, 2.5 µm. *n* = 6, 5, 4 for Con, DM-Sed, and DM-Ex groups, respectively. White boxes, fields with a zoomed-in view on the second row. A total of 6–10 images per animal were quantified in the following panels. (**B**–**E**) Quantifications of size, content, circularity, and cristae structure from TEM images. *, *p* < 0.05; **, *p* < 0.01; ***, *p* < 0.001, one-way ANOVA with post hoc multiple comparisons. (**F**) Representative mitochondrial consumption (OCR) curve from a control mouse. (**G**) OCR results. n.s., not significant. *, *p* < 0.05, one-way ANOVA with post hoc multiple comparisons at each respiratory condition. *n* = 7, 10, 11 for Con, DM-Sed, and DM-Ex groups, respectively. (**H**) Western blot assessment of protein carbonylation with quantification. **, *p* < 0.01, one-way ANOVA with post hoc multiple comparisons. *n* = 11, 11, 8. (**I**) Western blot assessment of mitochondrial proteins with quantification. mix, a mixture of mouse tissues serving as a technical control. No statistical significance was reported in all groups (one-way ANOVA).

**Figure 4 ijms-25-00633-f004:**
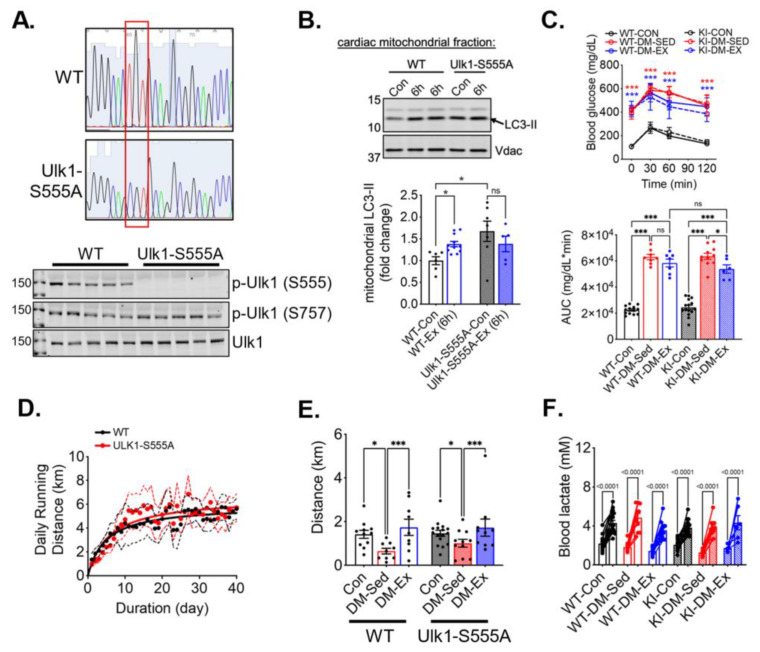
Ulk1 phosphorylation at S555 is required for acute exercise-induced mitophagy but not for exercise-mediated protection against exercise intolerance in diabetic mice. Top panel, the red box highlights the mutation of the S555 sequence. (**A**) Sequencing and western blot confirmation of ULK1-S555A generated by CRISPR/Cas9. (**B**) Exercise-induced mitophagy as indicated by western blot analyses of LC3 in centrifugation-isolated cardiac mitochondria. Ex (6 h), 6 h after an acute bout of exercise of 90 min treadmill running. n.s., not significant. (**C**) Glucose tolerance test of ULK1-S555A knock-in mice and their wildtype littermates after being subjected to the same diabetes and exercise intervention protocol. *n* = 7–15. n.s., not significant. ***: *p* < 0.0001 compared with control groups, two-way ANOVA with post hoc multiple comparisons. Some high data points were substituted with 650 mg/dL due to exceeding the upper limit of the glucometer. (**D**) The average daily running distance of the diabetic mice (WT-DM-Ex and KI-DM-Ex). The dotted lines denote the 95% confidence intervals of each group. (**E**,**F**) Treadmill running test results and blood lactate change before and after the running session to confirm exhaustion. *, *p* < 0.05; ***, *p* < 0.001, two-way ANOVA with post hoc multiple comparisons. *n* = 7–15.

**Figure 5 ijms-25-00633-f005:**
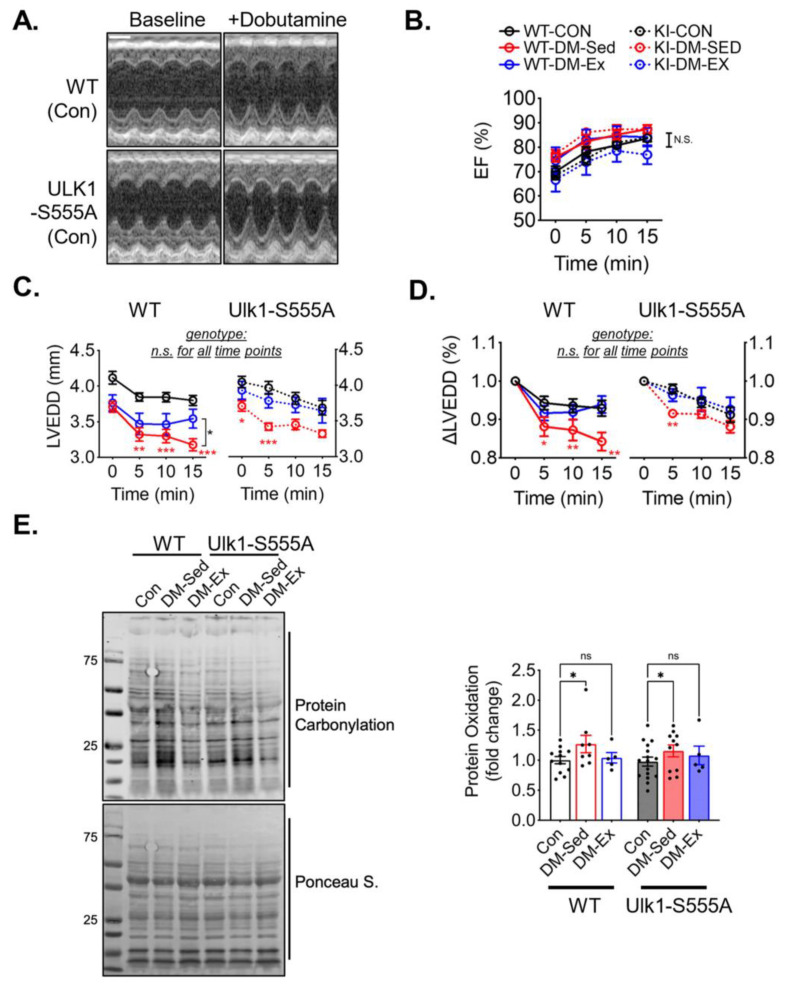
Ulk1-S555A mice retain exercise-mediated protection against diastolic dysfunction in diabetes. (**A**) Representative echocardiography images at baseline and 10 min post-i.p. injection of dobutamine (2.5 µg/g) of two control mice. Scale bar, 1 mm. (**B**–**D**) Ejection fraction (EF), LV end-diastolic diameter (LVEDD), and ∆LVEDD. n.s., not significant. *, *p* < 0.05, **: *p* < 0.01, ***: *p* < 0.001, two-way ANOVA with post hoc multiple comparisons at each time point. *n* = 7–15. (**E**) Western blot assessment of total protein carbonylation and quantification. n.s., not significant. *, *p* < 0.05, two-way ANOVA with post hoc multiple comparisons.

**Figure 6 ijms-25-00633-f006:**
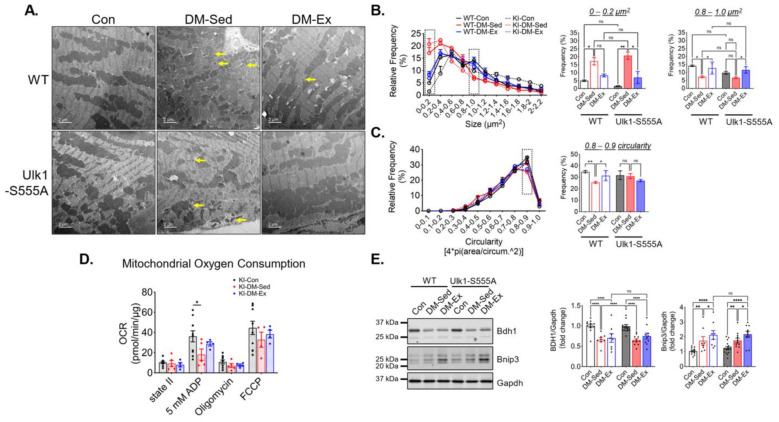
Ulk1-S555A mice retain exercise-mediated mitigation of mitochondrial abnormalities via alternative mechanisms. (**A**) Representative transmission electron microscopy (TEM) images of LV. Scale bar, 2 µm. *n* = 2. Yellow arrows, examples of very small (scattered) mitochondria. 5 randomly selected images from each mouse were quantified using FIJI. (**B**,**C**) Quantifications of size and circularity distributions TEM images. n.s., not significant. *, *p* < 0.05; **, *p* < 0.01, two-way ANOVA with post hoc multiple comparisons. (**D**) OCR results in the Ulk1-S555A mice. *, *p* < 0.05, one-way ANOVA with post hoc multiple comparisons. (**E**) Western blot assessment of heart lysates. *, *p* < 0.05; **, *p* < 0.01; ****, *p* < 0.0001, two-way ANOVA with post hoc multiple comparisons.

## Data Availability

The datasets generated during and/or analyzed during the current study are available from the corresponding author upon reasonable request.

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
