# Peer review of "Endurance Exercise Training Mitigates Diastolic Dysfunction in Diabetic Mice Independent of Phosphorylation of Ulk1 at S555"

_ijms, 2024, doi:10.3390/ijms25010633_

Round 1

Reviewer 1 Report

Comments and Suggestions for Authors

The work is interesting and shows important results, although the quality of the figures must be improved and some concepts revised.

1) Figure 1B, Y scale is very large and does not allow us to see the differences between the groups.

2) Figure 1D, 1E, 3B-I, 4, 5 and 6 review the text of the legends.

3) Some authors report that exercise allows controlling glucose levels and the glucose tolerance curve in diabetic models, which is not seen in Figure 1 D. Is there any explanation for this negative result?

4) Higher doses of oral glucose are normally used in the glucose tolerance curve than the one mentioned in line 406. Is there any reason to use a very low dose (200 mg/kg) and an I.P. route?

5) Figures 2A, 2B and 5A,  have a scale?

6) Figure 2C, Y scale is very large and does not allow us to see the differences between the groups.

7) In Figure 2E it seems that there are opposite results, since I would expect that with exercise there would be greater relaxation. Is there any special explanation for the results shown in Figure 2E?

8) In addition to the factors mentioned in your discussion, it is important to evaluate the excitability of the cardiac tissue in each of the conditions as well as the intracellular pH associated with metabolism and exercise.Although none of those factors are mentioned in the manuscript.

Reviewer 2 Report

Comments and Suggestions for Authors

In the paper titled "Endurance Exercise Training Mitigates Diastolic Dysfunction in Diabetic Mice Independent of Phosphorylation of Ulk1 at S555," the authors use a mouse model with diabetes to evaluate the hypothesis that endurance training can alleviate diastolic dysfunction by promoting mitophagy, specifically in the heart tissue. In this context, the authors focus on a mechanism mediated by the phosphorylation of Ulk1 at position S555. However, Guan, Y et al. describe that endurance exercise mitigates diastolic dysfunction in a diabetic mouse model but in an independent manner of the phosphorylation of Ulk1 at S555.

Although the results presented in the manuscript are interesting, minor revision is necessary to meet journal standards. The experimental quality of the manuscript is high, and addressing some minor issues will refine the content, ensuring that it conforms to the publication criteria. Authors are encouraged to make these adjustments to produce a version of the paper that meets the standards required for publication.

More specifically, the paper has minor issues.

Minor issues:

-Introduction Section: In the introduction, the authors should introduce the concept of autophagy, highlighting mitophagy as a specific type of autophagy. Additionally, it is crucial to mention that ULK1 is implicated in the initiation of the macroautophagy pathway. By incorporating these points into the introduction, the authors provide essential background information, enabling readers to grasp the significance of investigating mitophagy as a mechanism to preserve cardiac homeostasis in a diabetic mice model

Furthermore, it is advisable for the authors to discuss the dual role of mitophagy/autophagy in the context of cardiac alterations. This discussion can be integrated into either the introduction or the discussion section. By addressing the dual role, the authors will contribute to a better understanding for the reader, laying the foundation for comprehending the relationship between mitophagy and diastolic dysfunction in the context of diabetes.

-Due to the activation of ULK1 mediated by AMPK, I believe it would be interesting to investigate the status of AMPK and mTORC1 signaling. In this context, a Western blot analysis of AMPK and mTORC1 can shed light on this matter.

- I have a curiosity for the authors. In material and methods, the authors describe that "Heart sections were cut at 10 µm in O.C.T. using a Leica CM1860 cryostat (Leica 443 Biosystmes, IL, USA)", so I wonder if it is possible for the authors to perform histological gomori trichrome staining to characterize fibrosis in the ulk1 KI mouse model.

-According to my previous suggestion, I believe it would be beneficial to perform immunofluorescence analysis on cardiac tissue using an antibody against LC3B. This additional experiment could serve to confirm the status of autophagy in the Ulk1 knock-in mouse model. Immunofluorescence against LC3B would provide valuable visual confirmation of autophagic activity, complementing the Western blot data and strengthening the overall interpretation of the study results.
